# A Study on the Health-Related Issues and Behavior of Vietnamese Migrants Living in Japan: Developing Risk Communication in the Tuberculosis Response

**DOI:** 10.3390/ijerph20126150

**Published:** 2023-06-16

**Authors:** Sangnim Lee, Nhan Huu Thanh Nguyen, Shori Takaoka, An Dang Do, Yoshihisa Shirayama, Quy Pham Nguyen, Yusuke Akutsu, Jin Takasaki, Akihiro Ohkado

**Affiliations:** 1Department of Epidemiology and Clinical Research, The Research Institute of Tuberculosis, Japan Anti-Tuberculosis Association, Kiyose City 204-8533, Tokyo, Japan; 2Department of Global Health Research, Graduate School of Medicine, Juntendo University, Bunkyo-ku 113-8421, Tokyo, Japan; 3Department of Respiratory Medicine, National Center for Global Health and Medicine, Shinjuku-ku 162-8655, Tokyo, Japan; 4TB Action Network, Kiyose-shi 204-8533, Tokyo, Japan; 5Master’s Program in Global Leadership, Vietnam Japan University, Hanoi 77000, Vietnam; 6Department of Community and Global Health, Graduate School of Medicine, The University of Tokyo, Bunkyo-ku 113-0033, Tokyo, Japan; 7Department of Medical Oncology, Kyoto Miniren Central Hospital, Kyoto-shi 616-8147, Kyoto, Japan

**Keywords:** migrant, overseas-born, tuberculosis, health behavior, health-seeking behavior, risk communication, Vietnamese, Japan

## Abstract

Ensuring a healthy lifestyle for the increasing number of Vietnamese migrants living in Japan is a key public health issue, including infectious disease responses such as tuberculosis (TB). To develop risk communication in relation to the TB response, this study aimed to explore the health issues and health-related behaviors of Vietnamese migrants living in Japan using a mixed method. A survey was conducted on Vietnam-born migrants, aged 18 years and over, in Tokyo. The survey consisted of questions on the following components: (1) demographics; (2) health-related issues and behavior; and (3) health-seeking behavior, information, and communication. A total 165 participants participated in the survey. The majority of the participants were young adults. 13% of the participants responded that they were concerned about their health. Moreover, 22% and 7% of the participants reported weight loss and respiratory symptoms, respectively. 44% of the participants answered they had no one to consult about their health in Japan when they needed it, and 58% answered they had no awareness of any Vietnamese-language health consultation services. Logistic regression analysis revealed that people who contact family members living in Vietnam or overseas using social networking services (SNSs) when they needed to consult someone about their health (adjusted odds ratio (AOR) = 6.09, 95% confidence interval (CI) 1.52–24.43) were more likely to present with one or more of the typical TB symptoms, compared to those who did not consult someone in this manner. Current smokers (OR = 3.08, 95% CI 1.15–8.23) were more likely to have health problems compared to non-smokers. The key informant interviews revealed that individual factors, the health system, and socio-environmental factors may hinder Vietnamese migrants’ health-seeking and health-information-seeking behaviors in Japan. TB risk communication approaches for migrants need to be developed considering their health-related behaviors while addressing their health needs.

## 1. Introduction

In recent years, there have been noticeable changes in the demographics of foreign residents living in Japan. The major countries related to foreign residents living in Japan in 2012 were China (*n* = 652,555, 32%), South Korea and North Korea (*n* = 530,046, 26%), the Philippines (*n* = 202,974, 10%), and Brazil (*n* = 190,581, 9%). However, the number of Vietnamese residents has increased in the last decade (*n* = 52,364 (3%) in 2012; *n* = 432,934 (16%) in 2021) due to the rapid increase in the number of technical intern trainees and workers [1]. In 2020, Vietnamese people became the second largest group of foreign residents living in Japan following Chinese people [1]. The Vietnamese population also became the largest group of foreign workers in Japan in the same year [2]. These demographic changes in foreign residents have also impacted the epidemiology of infectious diseases in Japan in recent years [3]. One example is tuberculosis (TB).

The TB notification rate has been decreasing in Japan, reaching 9.2 per 100,000 population in 2021; then, Japan became a low-TB-burden country. On the other hand, the proportion of overseas-born cases (*n* = 1313) among all newly notified TB cases (*n* = 11,519) has increased, reaching 11.4% in 2021. Among the overseas-born newly notified TB cases, Vietnam-born cases became the largest group (*n* = 331) in 2019 and the second largest group in the years 2020 (*n* = 287) and 2021 (*n* = 263). It has been reported in the literature that overseas-born pulmonary TB patients had delayed access to TB treatment compared to Japanese patients [4]. Promoting migrants’ early access to healthcare and TB diagnosis has been a significant issue in the TB response and public health of Japan.

However, there are several challenges when approaching international migrants in terms of public health issues, including infectious disease responses. Migrants are likely to have limited access to health information in destination countries due to cultural differences and language barriers, socio-economic instability, and unfamiliarity with their access to the healthcare system [5]. In addition, health information published by the government may not reach migrant populations because of the limited understanding of migrants’ needs and social behaviors for information searching [6,7]. These mismatches can negatively affect infection prevention measures and might inhibit migrants from accessing healthcare services [8]. Given the mismatches between health program workers’ approaches and the needs of migrant populations, international organizations have stressed the importance of developing risk communication and community engagement (RCCM) with migrant communities during the COVID-19 pandemic [9]. This strategy should be fostered in TB responses for migrants, too, because TB also results in public health emergencies of mass infections in schools and workplaces [10,11]. It has been reported in the literature that public health emergencies, such as COVID-19, can affect TB detection and service delivery outcomes [12]. In order to develop effective risk communication strategies for migrants, it is necessary to understand the health-related thoughts, knowledge, and behaviors of the target population [6].

Under the circumstance that the number of Vietnamese migrants has increased in Japan in recent years, the International Organization for Migrants (IOM) conducted a study on healthcare access among Vietnamese technical intern trainees living in Japan during the COVID-19 pandemic in 2021 [7]. They identified several inhibiting factors concerning migrants’ access to relevant healthcare and information, such as financial, structural, and cognitive barriers. Several studies reported that Vietnamese migrants had difficulties in accessing medical services due to certain factors, such as language barriers, limited social support, and unfamiliar medical services procedures [7,13,14]. However, very few studies provided information about the health-related issues, concerns, and behaviors of Vietnamese migrants with diverse occupations and social positions in Japan from the migrants’ perspectives [15,16]. A better understanding of the broad health-related issues and behaviors of Vietnamese migrants needs to be considered when developing infectious disease responses, including risk communication approaches, in Japan. This process is essential to prevent mismatches from occurring between the needs of target populations and health programs. Thus, this study aims to explore the health issues and health-related behaviors of Vietnamese migrants. We also discuss the possible approaches to strengthening the TB response for Vietnamese migrants living in Japan.

## 2. Materials and Methods

### 2.1. Study Design

This study used both quantitative and qualitative methods. First, we conducted a survey on Vietnamese migrants. We then collected the necessary qualitative data based on key informant interviews to better understand the health-related issues and behaviors of Vietnamese migrants. This study involved participatory action research [17] and was implemented in collaboration with the Vietnamese migrant community throughout the entire research process. Human resources in the Vietnamese migrant community mainly included volunteer members of the TB Action Network. This network addresses the health and TB issues of Vietnamese migrants living in Japan, based on a migrant-centered approach. This non-profit network was established in the year 2020 during the COVID-19 pandemic between Vietnamese migrants, health professionals and researchers from Vietnam and Japan, the Vietnamese media, and individuals.

### 2.2. Quantitative Study

#### 2.2.1. Setting and Participants

A cross-sectional study was conducted on Vietnam-born migrants, aged 18 years and over, who attended a large-scale cultural exchange event held in Tokyo over a period of two days in June 2022. Researchers and volunteer members of the TB Action Network were involved in collecting data by directly inviting Vietnamese migrants who visited the event venue to participate in the study. The majority of the network members were Vietnamese migrants who could communicate with the participants in Vietnamese. Vietnamese migrants were informed about this study’s purpose, confidentiality, and their rights to withdraw from this study at any time prior to responding to the survey questions. Only those who provided consent to this study participated in the online survey using a smartphone. This process was written down on the online questionnaires, and informed consent was also obtained verbally. At the end of the survey, willing participants were invited to attend free health consultation services with medical interpreters, which was organized by the TB Action Network in collaboration with medical volunteers present at the event venue.

#### 2.2.2. Measures

The questionnaires were originally created in Japanese and then translated into Vietnamese. The pilot testing phase was conducted by two Vietnamese individuals and edited based on their feedback. The self-administered online questionnaire consisted of three components: (1) demographics; (2) health-related issues and behaviors; and (3) health-seeking behavior, information, and communication. There were 23 questions in total, with 1 additional question on women’s health for female participants only. No personally identifiable information was collected in this survey. Collected survey responses were stored in password-locked data sheets, with access only being provided to the researchers working for the data management team.

Our study attempted to determine the proportion of people who presented typical TB symptoms among Vietnamese migrant populations by asking the following questions: “Do any of the following symptoms persist for more than two weeks, including now (cough, phlegm, fever, difficulty with breathing, fatigue, none)?”, “Have you had a weight loss of 3 kg or more within a year (yes or no)?”, and “Have you ever had bloody phlegm (yes or no)?”.

In order to broadly understand the health-related behaviors of the study participants, we asked questions on various health-related behaviors, such as their history of COVID-19 vaccines and health check-ups, smoking, weekly alcohol consumption, and behaviors related to sexual and reproductive health. The component of “health-seeking behavior, information, and communication” was also included in the survey to consider developing a risk communication approach for Vietnamese migrants. Collected free-text answers provided for some questions on, for example, “Health information of interest” and “Most viewed Internet-based media to gather useful information for living in Japan”, were coded using the type of medical specialty or type of media.

In this study, we defined “health behavior” as any action undertaken by people who perceive themselves as healthy [18], “health-risk behavior” as any action that people undertake with a frequency or intensity of increasing risk of illness [19], and “health-seeking behavior” as any action undertaken by people who recognize that they have symptoms or health problems to seek healthcare assistance [20].

#### 2.2.3. Analysis

The main variables are presented in a frequency table using descriptive analysis.

In this study, we defined “having symptoms of TB” and “having problems about health” as the indicators for “health issues” in the statistical analysis. Chi-squared or Fisher’s exact tests were used to test the independence of variables related to participants’ characteristics using each indicator for health issues. Variables that may lead to the health issues defined as the indicators were selected, and a univariate logistic regression analysis was used to estimate the association between these variables and relevant health issues. A multiple logistic regression analysis was performed to estimate the odds ratio (OR) for each health issue. We included the data obtained from participants who responded to all survey components in this analysis. We excluded the data obtained from participants without a response to the demographic component in this analysis. Adjusted odds ratios (AORs) with a 95% confidence interval (95% CI) were also calculated. A *p*-value of less than 0.05 was considered statistically significant. All analyses were conducted using Stata version 16.1 (StataCorp, College Station, TX, USA).

### 2.3. Qualitative Study

#### Data Collection and Analysis

Semi-structured and in-depth interviews were conducted by Vietnam-born people with experience as volunteers assisting Vietnamese migrants with healthcare access, including language assistance in Japan. A convenience sampling method was used because such a human resource for Vietnamese migrants is very limited in Japan and also because of the limited study period. Three key informants were recruited and questioned about the health issues observed in Vietnamese migrants living in Japan, their perception about Vietnamese migrants’ health and health-seeking behaviors, and their experiences of supporting the health of Vietnamese migrants in Japan. The methodology for the key informant interviews is summarized in Table A1. Recorded interviews were transcribed verbatim, translated from Japanese into English, and then analyzed using inductive thematic analysis [21]. The data related to health-related issues and behavior were coded and carefully examined to identify the relevant themes. We removed personally identifiable information from the transcribed data during the data analysis process in order to ensure the anonymity of both interviewees and case examples.

## 3. Results

### 3.1. Quantitative Results

A total of 165 participants took part in the self-administered online questionnaire. Table 1 summarizes the demographic characteristics of the participants. Overall, 45% of them were in their twenties. By occupation, office workers (24%) and technical intern trainees (24%) accounted for almost half of the participants.

Table 2 illustrates the health-related issues and behaviors of the participants. Among the questions about typical TB symptoms, the highest proportion of responses corresponded to weight loss values of three kilograms or more within a year (22%). A total of 7% of the participants reported respiratory symptoms, while 33% reported having one or more TB symptoms. There were four participants who reported having more than four out of seven TB symptoms. Five people had a history of bloody sputum and two people experienced difficulties with breathing. Twenty-one people (13%) responded that they worried about their health. Among those, nine participants described their health concerns, such as backache, nosebleeds, herpes zoster, headache, fatigue, and allergic symptoms. The majority of the participants reported being non-smokers, with non-frequent drinkers who drink less than three days a week among them. About half of the survey participants reported that they use condoms when they have sexual intercourse. A total of 61% of the female respondents reported that they had never undergone cervical cancer screening tests in Japan; the main reason was that they were not aware of the procedure.

Table 3 illustrates the characteristics of health-seeking behavior and communication practices among the participants. Overall, 44% of them answered that they had no one to consult about their health status in Japan when they required assistance. Among the participants having someone to consult about their health status in Japan, the most frequent answer was “friends” (15%), followed by “family members and relatives living in Japan” (13%). Moreover, 58% answered that they had no knowledge of any health consultation services in the Vietnamese language in Japan. Facebook Messenger (49%) was observed to be the most popular communication tool that the study participants want to use when they want to consult a health support desk in the Vietnamese language in Japan. Although only one-fifth of the survey participants responded to the question on the health information they were interested in, women’s health was the most common response, followed by infectious diseases, including vaccines, and gastrointestinal-related information. The most popular Internet-based media type for migrants when collecting information on life in Japan was non-specific Facebook resources, followed by Vietnamese social media. For variables that may cause the health issues defined as indicators, the results of the chi-square test and Fisher’s exact test performed for each health issue indicator are described in Appendix A Table A2 and Table A3, respectively.

Table 4 presents the results obtained using the logistic regression analysis of the variables for the health issue indicator “having one or more symptoms of TB”. People who contact family members living in Vietnam or overseas using social networking services (SNSs) when they need to consult someone about their health (AORs = 6.09, 95% CI 1.52–24.43) were more likely to present with one or more symptoms of TB, compared to those who did not consult someone in this manner. Drinking for one day or more per week (AORs = 0.21, 95% CI 0.06–0.81) was identified as a protective factor for having TB symptoms compared with drinking less. Table 5 presents the results obtained using the logistic regression of the variables for the health issue indicator “having problems about health”. Current smokers (OR = 3.08, 95% CI 1.15–8.23) were more likely to have health problems compared to non-smokers in the univariate logistic regression analysis; however, this association was not statistically significant in the multiple logistic regression analysis. There were no statistically significant associations with other variables.

### 3.2. Qualitative Results

This study determined various health issues among Vietnamese migrants living in Japan reported in the interviews with the key informants. The qualitative analysis of the interviews generated three themes in particular regarding the health-seeking behavior of Vietnamese migrants living in Japan. These were “obstacles to migrants’ health seeking”, “vulnerability of recent young adult migrants”, and “communication in the SNS-based Vietnamese community”.

#### 3.2.1. Obstacles to Migrants’ Health-Seeking Behaviors

All interviewees emphasized the issue of language barriers when discussing the health issues often heard from Vietnamese migrants. In addition, the interviewees mentioned various issues related to the health-seeking behavior of Vietnamese migrants, including the issue of access to medical care arising from their lack of knowledge about the Japanese health system and health consultation services. An interviewee claimed she received personal requests from numerous female migrants, such as requests to accompany them to medical facilities and requests for medical interpretations over the phone.

##### Language Barriers

“The common problem is that they can only speak Vietnamese, so everybody contacts me. I have received so many inquiries such as serious women’s diseases, or … headache, or a lot of various stress and insomnia …”(Interviewee A)

Due to limited proficiency in the Japanese language, there were also consultation cases for patients who could not understand the physician’s explanation about TB, even after receiving a doctor’s examination. For an interviewee who is fluent in Japanese, the medical terminology was difficult.

“They first went to see a doctor on their own, and because they did not understand what the doctor said, they asked for our personal [consultation]. The Japanese word for tuberculosis is … because they only understand Vietnamese. Probably, Vietnamese people who are in their thirties do not know about tuberculosis. … Some time ago, I accompanied a patient together with an interpreter, but the language was difficult. … The doctor’s language … tuberculosis covers a broad area, and technical terms were difficult, so I studied.”(Interviewee A)

Regarding the public service of health screening, an interviewee claimed that female migrants with limited proficiency in the Japanese language are likely to miss opportunities for cervical cancer screening and other health screenings because they cannot read Japanese, even if public authorities mail coupons to them. She suggested that those announcements should be delivered in Vietnamese or written in simple Japanese.

##### Lack of Knowledge concerning the Japanese Health System and Limited Knowledge of Health Support Resources, in Addition to Language Barriers

An interviewee described the issue of poor health-seeking behavior among Vietnamese migrants. She also highlighted that even when Vietnamese migrants display some symptoms, they tend to give up or lose confidence in themselves, to the extent that they avoid seeking medical attention because of their limited ability to communicate in Japanese.

“When they have stomach trouble or something, Vietnamese people, in general, tend to leave it untreated, and do not consider going to hospital and to be detected any problem at early stage. They do not even know where to go or who to ask, and they all give up when they cannot communicate in Japanese on the spot.”(Interviewee A)

Recent Vietnamese migrants who understand neither the medical-related Japanese language nor the Japanese health system, would not know how to ask for help if they are in trouble. An interviewee claimed that, during the COVID-19 pandemic, recent migrants who had no knowledge about Japanese health systems, especially the medical care system, faced a considerable number of problems and did not know where to seek relevant, professional advice. Another interviewee suggested that regular training concerning healthcare access, such as disaster drills for local residents, would be necessary for migrants to cope with their health problems.

“They can speak Japanese, but they still don’t understand Japanese medical terms or the Japanese medical care system. … They find it difficult due to lack of confidence as they cannot communicate with the Japanese... They don’t know either how to call an ambulance when a friend falls ill. … In my city, evacuation drills organized by each municipality provide, for the sake of foreign residents, such information as how to evacuate....As with medical care, if possible, they would expect to know correct health information on where to contact, or where to seek advice from…, but the language is the biggest issue.”(Interviewee A)

Due to the social environment in their home country, Vietnamese migrants probably do not have much experience in seeking advice from specialists when they have mild health problems. Those with limited proficiency in Japanese believe that consultation services in Japan are only available in Japanese. An interviewee claimed that Vietnamese people did not know where they could obtain professional health advice and reassurance.

“I was able to learn various organizations of the local government [in Japan]. For example, they provide health checkups for babies … When I listen to my friends in Vietnam or something … there really isn’t much public medical support in Vietnam. … The [consultation] services provided by the government are quite limited. I think that Vietnamese people probably don’t know that there are such contact points through which they can always seek advice, or specialist contact points, in the town they live [in Japan].”(Interviewee B)

#### 3.2.2. Vulnerability of Recent Migrants When They Experience Health Problems

In addition to language barriers, the issue of access to medical care for recent migrants, such as international students, who have no one to turn to or talk to is also relevant. Additionally, migrants’ socio-economic conditions, such as being uninsured and undocumented, become a barrier to accessing healthcare services.

“International students are not managed by anyone, and some students are uninsured. They have not paid premiums for the national health insurance. Uninsured persons do not actually want to go to hospital very much. Their [symptoms] had become very serious by the time they went to hospital. Then there were a very small number of people whose visa expired and they ran away, and by the time they arrived at hospital their condition had become very serious—these cases are quite troublesome.”(Interviewee C)

#### 3.2.3. Communication in the SNS-Based Vietnamese Community

All interviewees discussed their support and engagement with Vietnamese migrants using SNSs, such as Facebook. An interviewee reported that Vietnamese migrants living in Japan tend to seek advice from their SNS communities instead of specialists. These migrants have very few people they can ask about access to healthcare services in Japan because they have few friends in Japan. “Can anyone chat with me now_?” is an SOS call.

“In the case of technical interns, the only persons they can talk to would be their supervisors. Then recently, everybody seeks advice through the Internet rather than by talking face-to-face with others. They send messages to this group, but … they only say, “Can anyone chat with me?” I instantly know that this person has a problem that he or she wants to share.”(Interviewee C)

As a promoting factor in health-seeking behavior, there is a requirement to improve the effective use of SNSs and public relations for migrants. There was an opinion emphasized by an interviewee that, while SNSs have the disadvantage of allowing incorrect information to be distributed, they are also necessary to provide correct information and guidance, leading to requests for specialist consultations. Other interviewees shared the idea of introducing information on TB with a short video and distributing it using the Internet. Another idea was to distribute health-access-related guidance to Vietnamese technical intern trainees through the supervising organizations after entering Japan.

“Probably, my biggest dissatisfaction or, an activity that I absolutely want to do, is to [distribute] correct information and [make] correct communication. In all the conversations exchanged between many [people] [on] Facebook, who should respond? For example, … there is a person who responds arbitrarily to an inquiry about herpes zoster..., but they aren’t a doctor...For example, ‘My stomach doesn’t recover even after medicine was prescribed by a doctor.’ Then there are [persons who] respond to this inquiry...‘You have nothing,’ … ‘Your illness is not something like folk, or fortune-telling in old times.’ I am quite unhappy with these, and what I really want is that someone distributes correct information on web pages.”(Interviewee A)

#### 3.2.4. Health Issues Observed in Vietnamese Migrants

Key informants reported a wide range of health issues observed in Vietnamese migrants living in Japan, based on their experience with assistance and daily observations. These included women’s health-related issues, respiratory symptoms, TB-related issues, symptoms caused by psychological stress, and headaches. Regarding respiratory symptoms, it was remarked that, because Japan has four distinct seasons with wildly varying weather conditions, Vietnamese migrants, including the interviewees, are prone to catch a cold after migrating to Japan. For some Vietnamese migrants, when they had a cold or bloody phlegm, they attempted to cure their symptoms using over-the-counter drugs. Another interviewee noted the poor health-risk perceptions of young adult male migrants in relation to asymptomatic cases and health problems that did not interfere with their daily lives. Interviewees noted the importance of addressing the requirements and consultations of numerous female migrants regarding women’s health, including gynecological issues and pregnancy. There was evidence of consultations with female technical interns who had strong fears of being threatened or forced to return to Vietnam by their company as a result of their pregnancy.

## 4. Discussion

In this study, the majority of the survey participants were young adults in their 20s. More than half of the participants were recent migrants who lived in Japan for less than four years. A range of health issues were identified among the survey participants. One-third of the participants presented typical TB symptoms, although a small proportion of the participants presented respiratory-related symptoms. Regarding the health-related behavior of Vietnamese migrants, we determined the positive aspects of “health behavior”, “health-seeking behavior” with an SNS preference, and “health risk behavior” that required some consideration. The survey also revealed the vulnerability of Vietnamese migrants in terms of social support if they experience health problems.

### 4.1. Health-Related Issues and Concerns Identified among Young Vietnamese Migrants

In the survey, some people reported experiencing respiratory symptoms, such as bloody sputum, and difficulties with breathing. Individuals with prolonged respiratory symptoms should be encouraged to seek early medical examinations and healthcare services. Care should be taken as there is a possibility that some people with mild respiratory symptoms may be coping with their symptoms using over-the-counter drugs, as was revealed in the key informant interviews. Approaches must be developed for people who may be reluctant to seek medical care unless their symptoms are severe.

Regarding the weight loss reported by 22% of the survey participants, several factors may be involved, particularly for people who lose weight without displaying any of the other symptoms. Since the majority of the participants were young adults who had recently immigrated to Japan, the potential causes of weight loss may include the physical and mental strain and stress caused by new living conditions, including the different food culture and working conditions in the host country. As explained by the key informants, recent immigrants, in particular, may have experienced stress and anxiety while living through the COVID-19 pandemic outside their home country. Several studies also reported the anxiety and psychological distress experienced by Vietnamese migrants in Japan during the COVID-19 pandemic [7,8]. In our survey, health concerns, such as headaches, backaches, and fatigue, were reported by the participants. These symptoms may have also been caused by physical or mental stress or work strain [22].

### 4.2. Health-Related Behavior

The majority of the survey participants experienced a health check-up in Japan in the last two years. In Japan, approximately 33% of overseas-born TB patients were detected as having TB during regular health check-ups, including health check-ups performed at school or work [4]. Chest X-rays are a well-known approach for the detection of TB, especially pulmonary TB [23]. However, there is a possibility that some participants were not included in the regular health check-up system at work or school, for example. The reasons for this may include their status, such as being unemployed or an overstayer. These migrants should be assisted in accessing free TB screening services run by several local governments.

Regarding the health-related behaviors of the survey participants, we observed that the majority of the participants presented healthy behavior, such as not smoking or drinking frequently. The result of the absence of frequent drinking among most Vietnamese migrants is consistent with a study conducted on Japanese-language school students in Tokyo [24]. Univariate logistic regression analysis showed that current smokers were more likely to worry about their health compared to non-smokers. Associations between smoking and health problems, such as lung diseases and TB infection, have been reported in the literature internationally [25]. The promotion of non-smoking campaigns is required to foster TB control measures. In our study, drinking once a day or more per week was a protective factor against “having one or more symptoms of TB”. However, a careful interpretation of the results is required for the following reasons. The major symptom displayed among typical TB symptoms reported by the survey participants was weight loss. Thus, there is a possibility that migrants with drinking habits were less likely to experience weight loss as a result. International studies reported that heavy drinkers were likely to experience weight gain [26]; however, additional examinations of our study participants are necessary. The COVID-19 vaccine was provided free of charge in Japan when this survey was conducted. The majority (59%) of the study participants received three doses or more of the COVID-19 vaccine. This rate was similar to the national average of Japan (approximately 60%) as of June 2022, when the present survey was conducted [27].

Regarding sexual health issues, three out of four sexually active people were practicing safe sex among the survey participants. Approximately 10% of female Vietnamese participants used the pill as a means of contraception. This result is similar to the trend in Vietnam and higher than the rate (approximately 3%) of pill use among Japanese women [28]. The pill is a contraceptive method that women can use by themselves; however, on the other hand, caution is required because this method does not prevent the occurrence of sexually transmitted infections (STIs). In recent years, the prevalence of STIs, such as syphilis, has increased rapidly in Japan, representing a significant public health concern [29]. Furthermore, the key informant interviews revealed challenges faced by pregnant female migrants who were concerned about maintaining a job or technical intern training. Most of the Vietnamese migrants we targeted were in their twenties considered to be sexually active, and at a high risk of contracting STIs and unwanted pregnancies without taking the appropriate measures. Given these conditions, in addition to raising awareness of TB issues, health education and counseling concerning reproductive and sexual health issues should also be addressed simultaneously [7].

In this survey, approximately 60% of the female Vietnamese participants had not undergone cervical cancer screening in Japan, which is operated in the municipality, for several reasons including a lack of knowledge of the procedures. Some Vietnamese women may have limited knowledge about this screening process because there is no national program for cervical cancer screening in Vietnam [30]. Cervical cancer screening rates for these participants were at the same level as the average screening rate (43.7%) in the last two years among women aged 20–69 years in Japan [31]. Further promotion of this screening knowledge is required in Japan.

### 4.3. Health-Seeking Behavior, Information, and Communication

Overall, 44% of the survey participants did not have someone to consult about their health when they needed to. This result suggests that there is limited social support for Vietnamese migrants living in Japan when they experience health problems. Poor family support and social networks would reduce migrants’ access to healthcare services [32]. In our study, among the participants who had someone to consult about their health issues, the highest proportion was evident for friends followed by family members; however, each proportion was not large.

People who consult their family members abroad using SNSs when they need to consult someone about their health issues were more likely to present some typical TB symptoms compared to people who do not consult someone in this manner. There are several possible reasons for this health-seeking behavior. First, many survey participants reported limited social support or limited knowledge of relevant resources to consult concerning their health issues in Japan. A key informant emphasized that migrants’ limited Japanese language proficiency may discourage them from seeking appropriate health advice from Japanese resources. Language barriers have been reported to be an obstacle to migrants accessing appropriate healthcare services [7,33,34,35] Under these circumstances, if a migrant experiences health problems, it is possible to consult with their overseas family using SNSs. On the other hand, those migrants may take a longer period of time to access healthcare in Japan, as they consult people living abroad who may have limited knowledge of the Japanese healthcare system. Most importantly, even if migrants displaying typical TB symptoms come into contact with someone first, they should be advised or supported to seek timely consultations with health personnel or at health facilities in Japan.

People from the same cultural and linguistic backgrounds can play an important role in encouraging other migrants to seek healthcare advice [36]. For example, the TB Action Network has supported access to healthcare services for Vietnamese migrants living in Japan upon referral by their family members living in Vietnam using Facebook Messenger [37]. In line with the other study results [15], Facebook Messenger was observed to be the most popular communication tool when the survey participants wanted to consult someone about their health issues in Japan. Nearly all the participants in our study used smartphones and were from the digital-use generation. Given the behavioral characteristics of Vietnamese migrants, improving health consultation services in Vietnamese using SNS-based messages can help migrants, especially those who may be overlooked, in their aim to access healthcare services in the early stages. An international study reported that Internet-based messages and texts are effective tools in promoting changes in health-related behavior [38]. Such SNS-based support services should be developed further in collaboration with migrant communities and facilitate collaboration with public authorities and formal expert groups in order to be trustworthy.

The majority of the participants did not have knowledge of the appropriate services to consult about their health issues in their native language in Japan. Our survey results imply that information available on health consultation services in the Vietnamese language may not be reaching Vietnamese migrants living in Japan sufficiently. Health consultation services for migrants are provided by several non-profit or public organizations in Japan. However, such services in the Vietnamese language are very limited, despite the fact that Vietnamese residents are the second-largest population among non-Japanese residents. The Organization for Technical Intern Training provides free consultations in the Vietnamese language for technical intern trainees to assist them with issues in their lives and training in Japan [39]. Trainees experiencing problems can consult the organization using phone or website-based messaging. Such official services are essential; however, most of these official or public services in Japan cannot be accessed using the SNS-based messaging services that many Vietnamese migrants in our survey and other studies feel comfortable using [7,15,40]. There are several benefits of instant messages using SNSs, such as being able to write inquiries at any time without requiring phone numbers or incurring phone charges.

A qualitative study found that individual factors, including life skills in the host country, the health system, and socio-environmental factors may hinder Vietnamese migrants’ health-seeking and health-information-seeking behaviors in Japan. These include the language barrier as a major factor, a lack of knowledge of Japan’s health system, young adult migrants’ social vulnerability in Japan, such as their limited social support when ill, and migrants’ life experiences in their home country. Language barriers can decrease migrants’ motivation and confidence to seek healthcare services in Japan, as was observed in the key informant interviews conducted in our study and other studies [7]. This may result in migrants having delayed access to [41] and ceasing to visit medical facilities. An effort must be made to break this vicious cycle at multiple levels, including individual, community, structural, and social levels.

The key informants emphasized that Vietnamese migrants, particularly young adult migrants and non-permanent residents who are recently socially unstable, were unable to receive social support when they experienced health problems. Efforts must be made to ensure that no migrants are left behind. In order to attempt to successfully identify migrants’ health needs and assist their health-seeking behavior in the early stages, it is necessary to consider the behavioral characteristics of Vietnamese migrants in the SNS-based Vietnamese community living in Japan. In line with the results obtained in previous studies, our study identified the issues experienced by Vietnamese migrants who did not know exactly where to go when they required medical services in Japan [7], recent Vietnamese migrants’ necessity for language assistance [24,40], and the need for someone to accompany Vietnamese technical intern trainees to the appropriate healthcare facilities when they experienced health problems [7,13].

For migrants, as one of the life skill programs, there is a requirement to improve health education programs that deal with adjusting to a new way of life when moving to a new host country, learning about the healthcare system, understanding how to deal with health problems, and exercising their right to ask for medical interpreters. This would enable migrants to better prepare for health problems in the future, including infectious diseases. These health education programs should be provided throughout the entire migration process, from the pre- [7] to post-migration stages [42].

With regard to infectious disease responses, the provision of public health information in their native language is essential to facilitate migrants’ health-seeking and health-information-seeking behaviors in a host country [7,15]. In addition to developing risk communication, there is a requirement to simultaneously develop a health support system that is culturally sensitive and migrant-friendly, including community health educators who assist migrants and nationwide public medical interpreter services [33,43], so that migrants with culturally and linguistically diverse backgrounds can gain smooth access to appropriate healthcare services in Japan.

### 4.4. What Is Required for the Development of Risk Communication in TB Responses for Vietnamese Migrants in Japan?


Risk communication concerning infectious diseases for Vietnamese migrants should address the TB issue not just alone but also together with other possible health-risk issues, such as STIs and other public health emergencies. It should also educate migrants by providing them with information concerning where to find appropriate health consultation services and information, and where and how to access healthcare services in the host country in the migrants’ own language.Interventions for risk communication must be fostered in both the pre-departure and post-migration processes, as well as online and offline.There is a need to strengthen collaboration between health and public authorities and the migrant community in order to deliver trustful information for TB and other health-risk challenges in a culturally and linguistically appropriate manner and to develop a support system for migrants’ easy access to healthcare services.Both risk communication and comprehensive health promotions are necessary to strengthen infectious disease responses and to address the health requirements of Vietnamese migrants living in Japan. Health promotion should facilitate regular health check-ups and address health issues, such as smoking, issues related to sexual and reproductive health, and other health issues caused by health risk behaviors. Furthermore, collaborations between health workers, community organizations, and policymakers can facilitate the development and implementation of comprehensive health programs to address the multifaceted needs of migrants identified in this study. By leveraging resources and expertise from various sectors, these collaborative efforts can lead to sustainable improvements in infectious disease responses, healthcare delivery, and health outcomes of the migrants.Vietnamese SNSs and instant-messaging tools provided by SNSs, such as Facebook Messenger, should be utilized as risk communication and health consultation channels for Vietnamese migrants living in Japan [37]. Further efforts are needed to develop these SNS-based, health-related support and consultation services and to make these resources trustworthy.


### 4.5. Limitations of This Study

Our study results should be interpreted with caution considering the following limitations. First, this study had a limited sample size. We needed to exclude the participants without a response to the demographic component from the multiple logistic regression analysis. In addition, the lack of occupational data limited the comparison of participants by occupation. Second, the survey sample was not necessarily representative of the Vietnamese migrant populations living in Japan, as the target audience was limited to visitors at an event taking place in Tokyo. Third, selection bias may have occurred due to the convenience sampling of both the survey participants and key informants for the interviews. However, for a study targeting hard-to-reach populations, such as international migrants, the convenience sampling method would provide relevant knowledge to better understand migrants [44]. In addition, some information and recall biases may have occurred because the participants’ responses to their health conditions, such as weight loss, relied on self-reporting. Moreover, social desirability bias could have also influenced the participants to provide responses they perceived as socially acceptable or favorable.

Despite these limitations, this study successfully documented the health status and related factors by approaching each of the migrants face-to-face and conducting the survey at a large-scale event venue in Tokyo. A migrant-centered approach enabled the participation of a number of Vietnamese people in this survey. Continuous research is needed to understand how to effectively deliver risk communication and health consultation service information to Vietnamese migrants.

## 5. Conclusions

This study provides new knowledge on the health status of Vietnamese migrants living in Japan, the majority of whom were recent young adult migrants. These migrants exhibited various health-related issues and concerns related to both infectious and non-infectious diseases, and they preferred to utilize SNSs for their health-seeking behavior. Public health workers and collaborators require a better understanding of such health needs and health-related behaviors of the target migrant populations in order to develop appropriate responses and risk communication approaches concerning infectious diseases, such as TB, for the migrants.

Our study had several limitations, such as a limited sample and selection bias. The survey sample was not necessarily representative of the Vietnamese migrant populations living in Japan, thus limiting the generalizability. This study used self-reported data, which may be subject to recall or social desirability biases. Despite these limitations, this participatory action research documented the health status and related factors successfully.

A strong collaboration with migrant communities is essential for fostering action research on the health of migrants. Further studies and the development of risk communication approaches in TB models would not only be helpful for the TB response but also for future public health emergencies. We believe that working with the Vietnamese community can pave the way for the development of an infectious disease response model for migrant populations.

## Figures and Tables

**Table 1 ijerph-20-06150-t001:** Demographics of Vietnamese people participating in the self-administered online questionnaire.

Category		*n*	%
Sex	Female	44	27
	Male	65	39
	Missing	56	34
Age group	18–19	1	1
	20–29	74	45
	30–39	30	18
	40–49	4	2
	Missing	56	34
Occupation or social positions	Students *	17	10
	Technical intern trainees	39	24
	Office workers	40	24
	Health personnel and long-term care workers **	6	4
	Self-employed or freelance/unemployed/others	7	4
	Missing	56	34
Prefecture of residents	Kantou area, including Tokyo ***	104	63
	Other areas	5	3
	Missing	56	34
Length of stay in Japan	Less than 2 years	20	12
	2–4 years	66	40
	5 years and over	23	14
	Missing	56	34

Notes: The number of subjects was 165. * Students include Japanese-language school students, students from other professional schools, and students from universities and higher education. ** Health personnel and long-term care workers include health personnel (medical doctors and nurses) and long-term care workers. *** The Kantou area includes Tokyo (47), Saitama (16), Chiba (14), Kanagawa (9), Tochigi (8), Ibaraki (7), and Gunma (3).

**Table 2 ijerph-20-06150-t002:** Health-related issues and behaviors.

Category		*n*	%
Symptoms lasting for a period over 2 weeks (multiple choices)	Cough	11	7
	Phlegm	11	7
	Fever	6	4
	Difficulties with breathing	2	1
	Fatigue	10	6
	Not applicable	139	84
Have you lost over 3 kg in body weight within a year?	Yes	36	22
	No	129	78
Have you ever had bloody phlegm?	Yes	5	3
	No	160	97
Are you worried about your health?	Yes	21	13
	No	144	87
If yes, please tell us what you are worried about (Free descriptions)	* Please see the footnote for answers		
COVID-19 vaccination status	Three or four doses	98	59
	Two doses	55	33
	Unvaccinated or vaccinated once	12	7
Do you have a health insurance?	Yes	162	98
	No	3	2
Health check-up history in Japan	Within two years (this year or last year)	138	84
	Two years ago or more/none/unknown	27	16
Place where you had the last health check-up in Japan	School	18	11
	Workplace (including workplace for technical intern trainees)	102	62
	Undergoing a health check-up at a place other than the workplace or school before starting work or attending school	29	18
	Never had a health check-up	7	4
	Others	9	6
Smoking	Smoke everyday	18	11
	Smoke not everyday	14	9
	Former smoker, but current non-smoker	18	11
	Non-smoker	115	70
Drinking per week	Drink more than three day	25	15
	Drink 1–2 days	14	9
	Drink < one day	55	33
	Non-drinker	71	43
Condom use for sexual intercourse	Use	77	47
	Not used with spouse because of marriage partners	16	10
	Unmarried but with trusted partner, so not used	7	4
	I do not purchase them because they are expensive	2	1
	Not applicable (e.g., no partner)	53	32
	Others	10	6
Contraceptives that you usually use (select all, including those used by your partner)	Male condom	85	52
	Pill	15	9
	IUS/IUD/implant	0	0
	Hormonal injection	2	1
	Not used	22	13
	Not applicable	36	22
	Others	17	9
Women-only: history of cervical cancer screening in Japan	Yes	17	39
	No	27	61
If your answer is no, please select the reason why (*n* = 27)	Because I received it in Vietnam	2	7
	Because it costs money	1	4
	I do not have time to attend the check-up (I am busy)	6	22
	I do not know of the procedure	7	26
	I do not know if I am the target	2	7
	I am embarrassed and scared	2	7
	It does not matter to me	2	7
	Others	2	7
	No answer	3	11

Abbreviations: IUS: intrauterine system; IUD: intrauterine device. Notes: The number of subjects was 165. * Among these, 9 participants described their health concerns as backache (*n* = 2), allergy rhinitis (*n* = 2), nosebleeds (*n* = 1), herpes zoster (*n* = 1), headache (*n* = 1), fatigue (*n* = 1), and facial skin allergy (*n* = 1).

**Table 3 ijerph-20-06150-t003:** Health-seeking behavior, information, and communication.

Category		*n*	%
Do you have anyone to consult about your health in Japan when you need it?	Yes	55	33
	No	72	44
	Missing	38	23
If yes, who do you consult?	Family members and relatives living in Japan	17	13
(Multiple answers allowed) (*n* = 55)	Family members and relatives living in Vietnam or abroad using SNSs	14	11
	School teachers and staff	2	2
	Work colleague/supervisors	10	8
	Supervising organizations for technical intern trainees	10	8
	Friends	19	15
	Support groups	5	4
	Others	17	13
Do you know of any support desks in Japan where you can consult about your health in the Vietnamese language when you need it?	Yes *	32	19
	No	95	58
	Missing	38	23
If yes, please write the name of the support desk (Indicates classified categories from free descriptions)	Number of valid responses *n* = 7 * Please see the footnote for answers		
When you want to consult anyone about your health in Japan, how would you prefer to contact a Vietnamese-speaking support desk/services? (Please select only one)	Facebook Messenger (chat function)	81	49
	Chat function of LINE	7	4
	Telephone	6	4
	E-mail	22	13
	Face to face	9	5
	Others	2	1
	Missing	38	23
What would you like to know/are interested in regarding health information? (indicates categories classified by medical specialty from free descriptions) Number of valid responses *n* = 31 (multiple answers allowed)	Women’s health-related issues	7	23
	Infectious diseases, including vaccines	5	16
	Gastroenterology-related issues	5	16
	Others **	19	61
Please tell us the name of your most frequently viewed (listened to) media sources (example: ○○ news, ○○ channel) on the Internet to receive the latest news necessary for living in Japan (indicates categories classified by medical specialty from free descriptions) (multiple answers allowed) Number of valid responses *n* = 17	Non-specific Facebook resources	10	59
	Vietnamese social media	7	41
	Japanese media	2	12
	Other social media platforms	2	12
Which of the following electronic devices do you own? (multiple answers allowed)	Smartphone	120	95
	Computer	40	2
	Tablet	11	9
	Others	3	2

Notes: The number of subjects was 165. * The responses include unnamed resources, such as labor unions (2), workplaces (1), schools (1), municipalities (1), and named resources, such as a hospital in Tokyo (1) and a non-profit organization that provides medical interpretation in Kanagawa prefecture (1). ** “Others” includes information related to eight medical specialties and two health-promotion-related sources.

**Table 4 ijerph-20-06150-t004:** Multiple logistic regression results for the potential factors associated with people having one or more symptoms of tuberculosis.

Category		Univariate				Multivariable AOR			
		OR	95% CI		*p*- Value	OR	95% CI		*p*- Value
Sex	Female	1.0				1.0			
	Male	1.36	0.59	3.16	0.47	2.19	0.78	6.10	0.14
Age group	Under 29	1.0				1.0			
	30+	0.89	0.37	2.14	0.79	0.66	0.24	1.80	0.42
COVID-19 vaccination status	Two doses or more	1.0				1.0			
	Unvaccinated or vaccinated once	0.38	0.08	1.79	0.22	0.21	0.02	2.13	0.18
Health check-up history in Japan	Within two years (this year or last year)	1.0				1.0			
	Two years ago or more/none/unknown	0.82	0.59	1.15	0.25	0.37	0.09	1.63	0.19
Smoking	Non-smoker *	1.0				1.0			
	Smoker (every day/not every day)	2.05	0.93	4.51	0.07	2.68	0.92	7.87	0.07
Drinking per week	Non-drinker or drinks for less than one day/week	1.0				1.0			
	Drinks for one day or more **	1.00	0.47	2.14	1.00	0.21	0.06	0.81	0.02
Do you have anyone to consult about your health status in Japan when you require it?	Yes	1.0				1.0			
	No	0.85	0.40	1.81	0.67	1.37	0.48	3.88	0.55
Consulting family and relatives living in Japan when I want to consult anyone about my health	No ***	1.0				1.0			
	Yes	1.27	0.43	3.73	0.66	1.99	0.41	9.65	0.40
Consulting family and relatives living in Vietnam or abroad (using SNSs) when I want to consult anyone about my health	No ***	1.0				1.0			
	Yes	3.53	1.13	10.99	0.03	6.09	1.52	24.43	0.01
Consulting friends when I want to consult anyone about my health	No ***	1.0				1.0			
	Yes	1.39	0.50	3.84	0.53	1.48	0.45	4.89	0.52

Multivariable models were adjusted for sex, age, and COVID-19 vaccination status. Abbreviations: AOR: adjusted odds ratio; CI: confidence interval; SNS: social networking service. Note: The number and percentage of people with the status of having one or more symptoms of tuberculosis are presented in Table A2. The number of subjects for the multiple logistic regression analysis is 109. * Non-smokers include those who have never smoked and former smokers who are currently non-smokers. ** “Drinks for one day or more” includes “drinks for 1–2 days” and “drinks for more than three days”. *** People who answered “No” include those who answered that they have no one to consult about their health in Japan when they need to.

**Table 5 ijerph-20-06150-t005:** Multiple logistic regression results for the potential factors associated with people who worry about their health.

Category		Univariate				Multivariable AOR			
		OR	95% CI		*p*- Value	OR	95% CI		*p*- Value
Sex	Female	1.0				1.0			
	Male	1.02	0.33	3.10	0.98	0.81	0.22	2.92	0.75
Age group	Under 29	1.0				1.0			
	30+	1.57	0.51	4.83	0.43	1.52	0.46	5.01	0.49
COVID-19 vaccination status	Two doses or more	1.0				1.0			
	Unvaccinated or vaccinated once	0.60	0.07	4.94	0.64	1.16	0.11	11.85	0.90
Health check-up history in Japan	Within two years (this year or last year)	1.0				1.0			
	Two years ago or more/none/unknown	1.24	0.38	4.02	0.72	1.35	0.31	5.86	0.69
Smoking	Non-smoker *	1.0				1.0			
	Smoker (every day/not every day)	3.08	1.15	8.23	0.03	3.07	0.90	10.47	0.07
Drinking per week	Non-drinker or drinks for less than one day/week	1.0				1.0			
	Drinks for one day or more **	1.75	0.65	4.70	0.27	1.02	0.25	4.20	0.98
Having one or more symptoms of TB	No	1.0				1.0			
	Yes	2.50	0.99	6.32	0.05	1.01	0.29	3.45	0.99
Do you have anyone to consult about your health in Japan when you require it?	Yes	1.0				1.0			
	No	1.32	0.45	3.88	0.62	1.43	0.44	4.58	0.55

Multivariable models were adjusted for sex, age, and COVID-19 vaccination status. Abbreviations: AOR: adjusted odds ratio; CI: confidence interval; TB: tuberculosis. Notes: The number and percentage of people who worry about their health are detailed in Table A3. The sub-categories for “Do you have anyone to consult about your health in Japan when you require it” were not included in this model because the logistic regression analysis did not work due to the sample size. * Non-smokers include those who have never smoked and former smokers who are currently non-smokers. ** “Drinks for one day or more” includes “drinks for 1–2 days” and “drinks for more than three days”.

## Data Availability

The data presented in this study are available upon reasonable request from the corresponding author (S.L.).

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
