# Peer review of "A Study on the Health-Related Issues and Behavior of Vietnamese Migrants Living in Japan: Developing Risk Communication in the Tuberculosis Response"

_ijerph, 2023, doi:10.3390/ijerph20126150_

Round 1

Reviewer 1 Report

General comments

Overall, the article would benefit from more clarity and focus. The introduction and methods sections could be improved by providing more detailed information on the study population and data collection process, including ethical considerations. The authors could also consider providing more detailed descriptions of the findings and more detailed interpretation of the implications for policy and practice. The conclusion could benefit from more clarity on the limitations of the study and its contribution to the field of TB research. Additionally, the article could be more concise, avoiding unnecessary background information and technical terms. The authors should also ensure compliance with the STROBE guidelines, including providing more detailed information on the statistical methods used and the representativeness of the sample. By addressing these areas, the article could be improved in terms of clarity, focus, and scientific accuracy, ultimately contributing to the field of TB research.

Introduction

The authors could provide more detailed information on the rationale for the study by explaining why examining the health-related issues and behaviors of Vietnamese migrants living in Japan is important. For example, the authors could discuss the unique challenges faced by this population, such as language barriers, cultural differences, and difficulties accessing healthcare services.

Methods

The authors could provide more clarity on the ethical considerations by stating whether they obtained informed consent from the participants and how they protected their privacy and confidentiality. For example, the authors could describe any measures taken to ensure that the survey responses were anonymous and how they stored and analyzed the data.

Results

The authors could include more detailed information on the associations between the variables by presenting correlation coefficients or regression coefficients. For example, the authors could state the magnitude and direction of the associations between the various health-related issues and behaviors examined in the study.

Discussion

The authors could provide more information on the potential implications of the findings for policy and practice by suggesting specific interventions or programs that could address the health-related issues identified. For example, the authors could discuss the potential benefits of providing culturally sensitive healthcare services or educational campaigns to raise awareness about TB and other health issues.

Conclusion

The authors could provide more clarity on the limitations of the study and how they may affect the generalizability of the findings by discussing potential sources of bias or confounding that may have influenced the results. For example, the authors could acknowledge that the study relied on self-reported data, which may be subject to recall bias or social desirability bias.

Overall

The article could benefit from more detailed descriptions of the methods and findings, including a more comprehensive discussion of the associations between the variables examined in the study. Additionally, the authors could consider providing more specific recommendations for future research and policy interventions.

There are some corrections that should be made. The authors might consider rewriting some sentences. Some abbreviations need to be clearly spelt out at first use. Grammar, punctuations etc need to be revised

Reviewer 2 Report

This paper is of some interest however it is not written that clearly and is a rather repetitive.

In lines 43 to 53 

please give  numbers and % for the different groups of foreign workers.  I have no idea if there are 1000 or 100,000 South Korea, China, Brazil, and the Philippines Vietnamese etc. working in Japan

What is the difference between foreign workers and foreign residents?

In line 56 there is no indication of the total number of cases of TB and so I have no idea if we are talking of 10 or 10,000 cases

It is not clear how the survey was carried out it appears it was carried data was collected online but also volunteers were used to collect data. This needs to be made clearer

Line 122 "three components: demographics; health-related issues and behaviors; and health-seeking behavior, information, and communication. "

The three components needed to be numbered as it is not very clear how these five topics break into three

There are four appendices. I am unclear if the information in these is essential to my understanding of the paper. My understanding is that any data that is essential for the reader to understand the paper should be included in tables. Only data that is of interest to those who require more detailed information should be put in an appendix.  This needs careful consideration.

Table 1 has 109 participants Table 2 has 165 participants, table 3 127 participants.

This is not the normal way of reporting data. I would expect all three tables to have 165 participants and that if there is a missing data it is recorded as such in a line in the table.

In table 2 Condom use for sexual intercourse 

And Contraceptives that you usually use (select all, including those used by your partner) 

One has 53 not applicable and the other 35 – which doesn’t makes sense, so how well was the questionnaire understood

In table 3 again there is a large number where there are no valid responses. The number in each cell should still total 165 and a line saying no response should make up the total

Line 188 heavy drinking is not defined and in table three there is no definition of a drink

The notes at the bottom of the tables are more detailed and they need to be. You don’t need to give the breakdown of three or four doses as it is not relevant. This applies to many of the other notes as well.  “For “three doses”, there are 96 subjects. For 220 “four doses”, there are 2 subjects”

In the tables the percentages could be rounded two whole numbers rather than to 1 decimal place.

It seems unlikely that drinking more alcohol is protective but this is not discussed.

Many of the quotes e.g. at line 371 are unnecessarily long and could be made clearer by being shorter

The discussion is weak and needs to be clearer

I do not know the literature in this topic but there must be many other migrant communities which have communication difficulties in relation to access to health care in the host country. There are very few references to other similar studies

This paper is of some interest however it is not written that clearly and is a rather repetitive.

The English is not bad and I fully accept that the authors are writing in the second or third language. However some of it needs checked.  For example  prohibiting” is use instead of “inhibiting”

I am not clear what this sentence means at line 100 “Human resources in the Vietnamese migrant community mainly include the volunteer members of the TB Action Network working for Vietnamese migrants in Japan concerning their health and TB based on a migrant-centered approach.”

Line 330 “She remarked that Vietnamese people do not know people who, or contact points through which, they can ask for advice with peace of mind” . This could be rewritten “She said that Vietnamese people did not know where they could obtain health advice and reassurance” which is shorter and clearer.  

The paragraphs are very long which makes it harder to read eg one paragraph from 483 to 509

Reviewer 3 Report

The manuscript entitled "A Study on Health-related Issues and Behavior of Vietnamese Migrants Living in Japan: for Developing Risk Communication in The Tuberculosis Response" reports the findings of a survey related to health issues conducted with Vietnam-born migrants in Japan.

-Title: The study covers various health issues but the title reflects as if the study was conducted for TB. Therefore, I recommend the authors to revise the title. 

- Line 31: "SNS" should be defined in the title.

- Lines 56-62: The general situation about TB in Japan should be added.

Instead of the term "Vietnamese-born", the authors should use "Vietnam-born", "born in Vietnam" or "Vietnamese".

Round 2

Reviewer 1 Report

None

None

Author Response

We were informed by the Assistant Editor that the second review report from reviewer 1 is "accept". reviewer 1 didn't have the comment in the second review.

Reviewer 2 Report

This paper has been significantly improved, but still needs some more work.

How many cases are Vietnam-born? I am not licenser of the size of the problem.

Line 59 Of the 1313 Among the overseas-born newly notified TB cases,  Vietnam-born cases became the largest group in 2019 and the second largest group in the  years 2020 and 2021

Line118 visited the event site 

event site could be an internet site but I think you mean the event venue?

Change the title Table 1. Demographics of Vietnamese people participating in the self-administered online questionnaire 

This is the most important change needed 

Table 1 and 3

The % should include the missing data.  If you say 60% of respondents were women that is not correct, as 56 are unknown/missing. So the number of women responding is between 65 and 121.  

N= 165

n               %

44              40 27

65              60  39

Missing                    56            34

This applies to all the lines and then the results being quoted in the text

In the discussion you need to be careful about using the % from the table 1 and table 3 as you can’t say that the % of the 109 apply to the 165.

  1. Line 627 First, this study had a limited sample. In what way is it a limited sample

Line 628 We needed to exclude the participants  without a response to the demographic component from the multiple logistic regression  analysis.  I am not clear why you excluded them and it is not described in the methods
